# On the Formalization of Constrained Adversarial Attacks in Network Traffic Analysis

**Anastasiya Nikolskaya & Dmitry Rybolovlev**
Ivannikov Institute for System Programming of the Russian Academy of Sciences (ISP RAS)
Moscow, 109004, Russia
Orel State University named after I.S. Turgenev
Orel, 302026, Russia
{nikolskaya.a.g,dmitrij-rybolovlev}@yandex.ru

## Abstract

This paper explores the challenges in the formalization of constrained adversarial attacks against machine learning-based intrusion detection systems for network traffic analysis. While effective at detecting anomalies and zero-day attacks in network traffic, such systems are vulnerable to adversarial attacks, particularly evasion attacks that introduce carefully designed perturbations into input data to cause misclassification. The structured and heterogeneous nature of network traffic and the complexity of extracted traffic features complicate the generation of functionally valid adversarial examples that represent realistic attack scenarios. Crafting realistic adversarial examples requires incorporating domain-specific constraints into the attack. This transforms the underlying optimization problem into a constrained optimization problem whose solution is non-trivial and does not guarantee convergence. This paper reviews existing approaches to incorporating domain constraints, discusses the limitations of traditional distance metrics in the context of network traffic, and highlights challenges arising from differences between feature-space and traffic-space attacks. We conclude that domain constraints must be accounted for as comprehensively as possible while preserving practical applicability, and that the effectiveness of constrained attack methods and constraint-handling approaches remains an open question.

## 1 Introduction

Network traffic analysis is an area in which machine learning (ML) methods are extensively applied. For example, intrusion detection systems (IDS), which are designed to detect cyberattacks in network traffic, can employ ML methods to detect threats that traditional signature-based IDSs cannot handle. While effective at detecting zero-day attacks and traffic anomalies that may indicate malicious network activity, ML-based systems are vulnerable to adversarial attacks.

Of particular importance are adversarial evasion attacks. Evasion attacks aim to mislead a trained model into making incorrect predictions, for example, causing the victim IDS to misclassify malicious network traffic as benign. To achieve this, these attacks introduce carefully crafted perturbations into input data, aiming to make the resulting adversarial examples cross the target classifier's decision boundary.

In order to be effective, adversarial examples generated by an evasion attack against ML-based IDS must be misclassified by the model as benign, while still preserving their malicious functionality and representing genuine, legitimate network traffic. However, network traffic is heterogeneous and complex: it consists of packets with a specific structure determined by different network protocols. Moreover, an IDS typically classifies network traffic based on the network traffic features extracted from it and not on raw traffic itself. These features may have different types, sets of admissible values, and interdependencies among them. All these factors constrain the feasibility of the applied adversarial perturbations.

The classical (unconstrained) adversarial attacks generally introduce arbitrary perturbations, which may violate inherent network traffic constraints and make the crafted adversarial examples physically infeasible. Consequently, adversarial examples generated without considering domain-specific constraints may fail to represent realistic attack scenarios. Properly accounting for domain constraints is necessary to ensure that generated adversarial examples remain consistent with feasible network behavior and can potentially be reproduced in real-world environments. These limitations motivate the investigation of adversarial evasion attacks that explicitly incorporate domain-specific constraints.

Currently, constrained adversarial attacks remain insufficiently explored. Of particular importance is the problem of rigorously formalizing a constrained adversarial attack in a way that would enable the effective and comprehensive incorporation of network traffic domain constraints. In such a constrained domain, the traditional mathematical formulation of an evasion attack may lead to a non-trivial optimization problem with no guarantee of convergence. Addressing these challenges is necessary for constructing realistic evasion attacks, which in turn may enable a more accurate evaluation of IDS robustness and facilitate the development of more effective defense mechanisms.

The main contributions of this paper are as follows:

- We analyze existing approaches for the incorporation of domain constraints and highlight their key limitations.
- We examine the applicability of traditional distance metrics for assessing the imperceptibility of adversarial perturbations in network traffic.
- We explore the challenges arising from the dual nature of network traffic processing, considering both extracted features and raw traffic data.
- We investigate the feasibility of fully incorporating all domain constraints.

## 2   DOMAIN CONSTRAINTS IN NETWORK TRAFFIC ANALYSIS

Network traffic features represent different characteristics of network packets or network flows, for example, statistical characteristics of the processed traffic. Typically, network traffic features are represented in the form of tabular data. The exact set of features depends on the specific feature extraction tool used. For example, features may include the protocol number or the average length of the data field of forward TCP/IP packets.

Extracted network traffic features generally have different ranges or sets of admissible values. For example, numerical statistical features are represented by real values (e.g. data flow rate or Flow Bytes/s) or integer values (e.g. total number of bytes or TotBytes). Different flags are typically binary, while categorical features such as network protocols (e.g. UDP, TCP) may be encoded as string values, binary values, or integer values.

As noted earlier, an adversarial example in the network traffic analysis domain must preserve its effectiveness. Wang et al. (2023) refer to this preservation of network traffic meaningfulness and its original functionality as preserving the intrinsic property.

When unconstrained adversarial attacks are used, the network traffic processed by IDS may lose its functionality due to the violation of the underlying network protocol. Inherent constraints include not only admissible ranges or sets of admissible feature values but also, for example, existing dependencies among features or the requirement of feature immutability.

Table 1 presents examples of violations of specific domain constraints which can occur during an unconstrained adversarial attack.

These specifics of network traffic processing and, consequently, the need to integrate domain constraints in IDS classifier design complicate the formalization of constrained adversarial attacks.

## 3   EVASION ATTACKS AS OPTIMIZATION PROBLEMS

Adversarial attacks originated in image processing (Szegedy et al., 2013), and the resulting formulations and methods were subsequently adapted to other domains. A mathematically rigorous

Table 1: Examples of domain constraint violations. Here, $x_i$ denotes a feature of the original example, and $x_i'$ denotes the corresponding feature of the adversarial example.

| CONSTRAINT TYPE | CONSTRAINT | EXAMPLE OF VIOLATION |
|---|---|---|
| Domain of admissible values of a non-negative feature | $x_1' \geq 0$ | $AveragePacketSize = -1$ (average packet length cannot be negative) |
| Domain of admissible values of an integer feature | $x_1' \in \mathbb{Z}$ | $MaxPacketSize = 0.5$ (maximum packet length in a network flow cannot take a real value) |
| Mutually exclusive binary features | $x_1' \neq x_2',$ $x_1', x_2' \in \{0,1\}$ | $SYN$ and $FIN$ flags are set to 1 (the SYN and FIN flags, used to establish and terminate a network connection, respectively, cannot be set simultaneously according to the TCP/IP protocol) |
| Dependency exists | $x_1' \leq x_2'$ | $AveragePacketSize > MTU$ (practically unattainable: the Maximum Transmission Unit (MTU) is a network device setting that limits the packet size transmissible without fragmentation; thus, any packets exceeding this threshold are fragmented) |
| Dependency direction (inverse dependency) | $x_1' < x_1 \Rightarrow$ $x_2' > x_2$ | When $IAT$ decreases, the $FlowBytes/s$ feature also decreases (in practice, when the inter-arrival time (IAT) decreases, the transmission rate usually increases) |
| Dependency among several features | $x_1' = x_2'/x_3'$ | $FlowBytes/s \neq TotBytes/FlowDuration$ (the data transfer rate is defined as the ratio of transmitted bytes to the flow duration) |

formalization of an evasion attack expresses it as an optimization problem that seeks to minimize a distance metric between the original example and the adversarial example derived from it.

For instance, Carlini & Wagner (2017) formulate the optimization problem for generating adversarial images as follows:

$$\begin{aligned} \text{minimize} \quad & D(x, x + \delta) \\ \text{such that} \quad & C(x + \delta) = t \\ & x + \delta \in [0, 1]^n. \end{aligned} \tag{1}$$

Here, $x$ denotes the original image, $\delta$ is an adversarial perturbation, $D$ is a distance metric, $t$ is the target label, $n$ is the number of input dimensions (pixels in the image), and $C(x + \delta)$ is the label predicted by the classifier for the adversarial example.

In such formulations, a distance metric quantifies the similarity between the original example and its adversarial counterpart, essentially reflecting the imperceptibility of the applied perturbations. $L_p$ norms ($p = 0, 1, 2, \infty$) are widely adopted as distance metrics, with the choice of a particular norm depending on the application domain and data characteristics. For images, the $L_2$ norm is often employed. For heterogeneous data, particularly tabular data, the $L_0$ norm is especially relevant, as it minimizes the number of modified features (Mathov et al., 2022). In the context of network traffic classification, the applicability of these metrics is questionable, since they do not account for feature interdependencies. Moreover, in general, imperceptibility of perturbations to human observers is not required, unlike in the case of images or tabular data intended for manual expert analysis. Cortellazzi et al. (2025) argue for using high-confidence attacks rather than attacks based on minimal perturbation. High-confidence attacks aim to find adversarial examples for which the classifier predicts an incorrect label with confidence higher than for any other class (Carlini & Wagner, 2017).

Overall, incorporating domain constraints requires introducing additional conditions into the optimization problem, resulting in a constrained optimization problem. However, solving it may be non-trivial and not guarantee convergence, for example, because of non-convex or nonlinear constraints, or non-differentiable functions (Mathov et al., 2022; Cortellazzi et al., 2025).

## 4 INCORPORATING DOMAIN CONSTRAINTS INTO OPTIMIZATION PROBLEMS

A unified methodology for incorporating domain constraints into the adversarial example search problem has not yet been developed. To support this claim, we systematically analyzed 16 studies published between 2020 and 2025 that propose methods for constrained adversarial attacks, primarily in the network traffic domain. Our analysis reveals two common approaches to incorporating domain constraints into adversarial example generation.

One approach is to modify an optimization procedure used to generate an adversarial example in a classical attack. For example, in the C-IFGSM method (a modification of IFGSM (Kurakin et al., 2016)), in order to obtain an adversarial example while preserving certain types of feature correlations, the Hadamard product between the constraint matrix and the matrix of gradient signs is used, which provides control over the direction and magnitude of the introduced perturbations (Tian et al., 2020):

$$X_0^{adv} = X,$$
$$X_{N+1}^{adv} = Clip_{X,\varepsilon}\{X_N^{adv} + \alpha C_n \odot sign(\nabla_X J(X_N^{adv}, Y_{true}))\}. \tag{2}$$

In their formulation, $X_N^{adv}$ denotes the adversarial example generated at the $N$-th iteration from the original example $X$, $J(X, Y_{true})$ is the cost function; $\varepsilon$ is the largest size of adversarial perturbation; $Clip_{X,\varepsilon}(X^{adv})$ denotes the element-wise clipping function; $\alpha$ is the single step size; $C_n$ is the constraint matrix; and $\odot$ is the Hadamard product.

However, the authors note that only certain types of correlations are considered in order to reduce data analysis time, and the identified dependencies between features may not hold for all data points. In addition, the constraint matrix is constructed for feature pairs, which limits the ability to account for complex dependencies among features.

Note that this attack definition contains the clipping function ($Clip$). Such functions preserve ranges of admissible feature values and have been employed in several studies (Simonetto et al., 2022; Simidžioski, 2021). In the above formulation (Tian et al., 2020), the clipping function truncates feature values $X_i^{adv}$ that fall outside the interval $[X_i - \varepsilon, X_i + \varepsilon]$ to the nearest boundary.

Another example of the first approach is the C-PGD attack (a modification of PGD (Madry et al., 2017)). This attack employs a recovery function that updates the features at each iteration in order to restore constraints (Simonetto et al., 2024):

$$x^{(k+1)} = R_\Omega(P_s(x^{(k)} + \eta^{(k)}\nabla\mathcal{L}(x^{(k)}, y, h, \Omega))). \tag{3}$$

In their formulation, $x^0 = x_{orig}$ (the original input), $P_s$ denotes the projection onto $\mathcal{S} = \{x \in \mathbb{R}^d, ||x - x_{orig}||_p \le \epsilon\}$, $\epsilon$ is the maximum perturbation threshold, $\nabla\mathcal{L}$ is the gradient of the custom loss function $\mathcal{L}$, and $\Omega$ is the constraint set.

However, the authors demonstrate that this attack can be mitigated by additional manually constructed non-convex constraints on pairs of the most important mutable features.

Note that projection is an important concept that is also used in other studies (Mathov et al., 2022; Simonetto et al., 2022). This operation projects the adversarial example from the original search space into a new constrained space (feasible space), thereby enforcing constraint satisfaction for the adversarial example, i.e., ensuring its feasibility.

The loss function of C-PGD is an example of another common approach to incorporating domain constraints – namely, a modification of the loss functions or the objective functions used. This approach can be applied simultaneously with the first approach and typically involves using penalty methods. For example, the loss function in C-PGD is defined as follows (Simonetto et al., 2024):

$$\mathcal{L}(x, y, h, \Omega) = l(h(x), y) - \sum_{\omega_i \in \Omega} penalty(x, \omega_i). \tag{4}$$

Here, $x$ denotes the input example, $y$ is the correct label, $l$ is the basic loss function, $h$ is the classifier, and $penalty$ is the penalty function that measures the violation of the constraint $w_i$.

This approach is generally more versatile than modifying a specific attack, as introduced functions can be reused in other attacks. For example, in addition to the penalty functions for violating domain constraints in C-PGD, Simonetto et al. (2022) propose using the same penalty functions in the objective functions of the multi-objective genetic algorithm MoEvA2.

However, the addition of penalty terms may lead to convergence issues under a large constraint set (Simonetto et al., 2022). Moreover, a penalty term can be multiplied by a coefficient that controls the relative importance of the constraints. This penalty coefficient is typically tuned empirically, which further complicates the successful generation of adversarial examples.

An important limitation of the considered attacks, as well as many other popular attacks designed for neural networks, is their reliance on the gradient of the loss function. This restricts their applicability to models with non-differentiable loss functions, such as Random Forests.

## 5 ATTACKS IN TRAFFIC SPACE

It is important to note that adversarial attacks may introduce perturbations either into extracted network traffic features (feature-space attacks) or directly into network traffic packets. The latter are referred to as problem-space attacks (Wang et al., 2023) or traffic-space attacks (Simidžioski, 2021). However, traditional inverse mapping techniques are ineffective for reconstructing network traffic packets from extracted features, since this mapping is neither invertible nor differentiable (Wang et al., 2023). The use of flow-based features with latent dependencies further increases the computational complexity of reproducing perturbations in real network traffic (Hore et al., 2025; Han et al., 2021). Therefore, generating network traffic from a feature-space adversarial example is generally a complex computational task.

It is also important to consider that an IDS may use classical ML models effective for tabular data, such as Random Forests. As noted above, this limits the applicability of common gradient-based attack methods. For such models, one must use either specialized methods capable of handling non-differentiable loss functions or substitute models exploiting the transferability of adversarial examples. However, the success of the latter approach may vary (Levy et al., 2024; Grini et al., 2025).

Traffic-space attacks are constrained by the use of permissible packet modifications. In Hore et al. (2025), the problem of searching for constrained adversarial examples in traffic space is presented as a sequential decision-making problem formulated as a Markov Decision Process and solved using deep reinforcement learning:

$$R_t = r_t + \gamma Q(s_{t+1}, \arg\max_a Q(s_{t+1}, a; \theta_t); \theta'_t). \tag{5}$$

Their formulation is based on the Double DQN (DDQN) algorithm introduced by van Hasselt et al. (2016), with some changes in notation. Here, $R_t$ is the target value; $r_t$ is the reward at the time step $t$; $Q$ is the action-value function; $\gamma$ is the discount factor; $s_{t+1}$ is the next state of the environment; $\theta_t$ and $\theta'_t$ are the parameters of the policy and target networks, respectively; and $a$ represents the perturbations, namely permissible packet modifications, such as increasing the maximum segment size (MSS) or decreasing the window size in bytes.

However, this attack has limited practical applicability because it uses only packets transmitted in the forward direction (from host to server), whereas many attacks, especially TCP-based attacks, depend on bidirectional dynamics. Moreover, on average, 45% of successful adversarial examples for all models tested in the study were out-of-distribution (OOD) examples. This implies that the effectiveness of the attack may significantly decrease when OOD detection methods are applied.

Han et al. (2021) formulated an unconstrained optimization problem by defining safe traffic mutation operations and a traffic reconstruction function that reconstructs network traffic from packet meta-information vectors:

$$\arg\min_x \mathcal{L}\big(\mathcal{E}'(\mathcal{R}(x)), f^*\big). \tag{6}$$

Here, $\mathcal{L}$ denotes the distance metric between two feature vectors, $\mathcal{R}$ represents the traffic reconstruction from packet meta-information vectors $x$, and $\mathcal{E}'$ is the surrogate feature extractor function.

However, the authors note that the resulting problem is a complex (NP-complete) combinatorial optimization problem due to the discrete nature of most dimensions in the meta-information vectors and the non-differentiability of $\mathcal{E}'$. Moreover, the proposed formulation does not take payload into account when generating packets, since the authors focus on non-payload-based IDSs. This approach excludes a large number of attacks that can be detected based on payload analysis, for example, SQL injection, which limits the practical applicability of the method.

## 6   Comprehensive Domain Constraint Incorporation

The full incorporation of domain constraints into an adversarial attack allows for the generation of the most realistic examples. However, as noted earlier, with a large number of constraints, such an attack may fail to produce a solution. For example, in an experiment with the C-PGD attack on network traffic from the CTU-13 dataset, a total of 360 constraints prevented generating even a single example satisfying them, since each penalty term contributed different, non-collinear, or even opposing gradients (Simonetto et al., 2022).

In high-dimensional feature spaces or domains with complex or numerous constraints, specifying all domain constraints is difficult. This is particularly challenging when interdependencies among features are present. Furthermore, attacks that consider only explicitly specified constraints may overlook hidden dependencies if such dependencies exist.

Moreover, network traffic may correspond to different protocols with varying packet structures, and even for the same protocol, the feature space may vary depending on the feature extraction tools used. This issue is further exacerbated by the fact that both packet-level and flow-level features can be employed.

These characteristics call into question the feasibility of fully accounting for all domain constraints.

## 7   Conclusions

In the field of network traffic analysis, traditional adversarial attacks may produce adversarial examples that violate inherent network traffic constraints, such as feature immutability or dependencies between features. Consequently, such adversarial examples cannot occur in a real-world network environment.

Constrained adversarial attacks take domain constraints into account and aim to enable the generation of realistic adversarial examples. However, the formalization of such constrained attacks as a constrained optimization problem poses notable challenges in practice, which can affect the effectiveness and scalability of various methods for incorporating domain constraints.

Our results indicate that the effectiveness of both these methods and the constrained attacks methods remains an open question. Classical gradient-based attacks may have limited applicability in network traffic analysis, as they can be applied directly only to models with differentiable loss functions. Attacks that use penalty methods may face optimization issues under a large number of constraints and, therefore, scale poorly to real-world IDSs. Traffic-space attacks may also have limited applicability in real-world conditions. Moreover, the use of traditional distance metrics may be inappropriate for the network traffic analysis domain.

Collectively, these limitations highlight the challenges of applying constrained adversarial attacks in practical network environments. However, when formalizing a constrained adversarial attack, it is necessary to account for domain constraints as comprehensively as possible while preserving the practical applicability of the method.

Future research should focus on developing more effective constrained adversarial attack methods, alongside rigorous formalization of these attacks to overcome existing limitations.

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
