# OpenReview forum: "On the Formalization of Constrained Adversarial Attacks in Network Traffic Analysis"
_mathai.club/MathAI/2026/Conference — 2026 Oral_

### Official Review · Reviewer_By11 · 2026-03-13
**Reject. It is purely a review and position paper.**

**Rating:** 5
**Confidence:** 2

**Review:**

Summary: The paper investigates the formalization of constrained adversarial attacks against machine learning-based Intrusion Detection Systems (IDS) for network traffic. It argues that standard adversarial attacks are infeasible in this domain because they violate network protocol constraints (e.g., flag combinations, feature dependencies). The authors categorize these constraints, analyze existing constrained attack methods (gradient-based and penalty-based), and highlight the limitations of traffic-space attacks. The paper concludes that incorporating the full set of domain constraints remains an open problem due to the non-trivial nature of the resulting constrained optimization landscape.

Strengths: Rigorous Problem Definition: The paper provides a clear and necessary formalization of the "constrained optimization" problem specific to network traffic, distinguishing it clearly from the image domain.
Critical Analysis: It offers a valuable critique of existing methods (like C-PGD and MoEvA2), correctly identifying why penalty methods struggle with large constraint sets (conflicting gradients, convergence issues) and why Lp norms fail to capture semantic similarity in this domain.

Weaknesses: Survey Nature: The paper lacks original results. It does not propose a new algorithm, prove a new theorem regarding the complexity of the problem, or provide experimental validation of a new hypothesis. It is purely a review and position paper.
Lack of Solution: While the paper successfully identifies the "gap" and formalizes the difficulties, it offers no theoretical or practical solution to the problems raised. It concludes by stating the problem is open, which limits its contribution to the state of the art.

Recommendation: It is a high-quality survey/position paper but lacks the novel theorems, proofs, or experimental advancements required for acceptance. It would be better suited for a journal special issue or a survey track.

---

> ### Author Rebuttal · Authors · 2026-03-13
>
> We thank the reviewer for the careful reading and for acknowledging the clarity of the problem formulation and the critical analysis of existing constrained attack methods.
>
> We agree that our work has elements of a position paper, which is intentional. The goal of the paper is not to propose yet another attack algorithm, but to explain why existing algorithms systematically fail in realistic network settings.
>
> While the paper reviews prior approaches, its purpose is to provide a critical analysis of existing formulations and their limitations in the presence of protocol constraints and feature dependencies. The main contribution of the paper lies in this synthesis and analysis, which highlights a structural mismatch between standard adversarial optimization frameworks and the requirements of network traffic modeling.
>
> We believe that identifying and clearly articulating this gap is an important step toward developing more realistic attack methodologies.
>
> The contribution of the paper is therefore primarily analytical.
>
> We thank the reviewer again for the helpful comments.

---

### Official Review · Reviewer_3YQy · 2026-03-13
**Weak Accept (Accept with Minor Revisions)**

**Rating:** 6
**Confidence:** 4

**Review:**

## Summary

The paper addresses the problem of formalizing constrained adversarial attacks against machine learning-based intrusion detection systems (IDS). The authors investigate a fundamental challenge: classical adversarial attacks developed for computer vision, when transferred to the network traffic domain, produce examples that violate inherent domain constraints such as semantic consistency, physical realizability, and functional validity of network packets.

The principal contribution of this work lies in providing a systematic survey of existing approaches to incorporating domain constraints into the optimization formulation of attacks, analyzing the limitations of conventional distance metrics (L<sub>p</sub>-norms) for heterogeneous tabular data, and examining the distinctions between feature-space and traffic-space attacks. The authors conclude that the effectiveness of existing constrained attack methods remains an open problem and that their scalability to real-world IDS raises serious concerns.

---

## Strengths

### Quality

The paper demonstrates a thorough understanding of the subject matter and provides a rigorous analysis of the relevant literature. The authors clearly articulate the mathematical formulation of the problem, drawing an analogy with the classical Carlini & Wagner (2017) formulation and systematically introducing constraints into the optimization objective. The survey of methods (C-IFGSM, C-PGD, reinforcement learning-based approaches) is conducted in a methodical and technically precise manner.

The catalogue of domain constraint violations presented in Table 1 is informative and effectively illustrates the non-trivial nature of the problem. The examples provided — negative packet size, simultaneous setting of SYN and FIN flags, violation of the FlowBytes/s ratio — convincingly demonstrate why unconstrained attacks are unsuitable for the network traffic domain.

### Originality

As a survey, the paper offers a valuable systematization of existing methods. Particularly noteworthy is the discussion of the distinction between feature-space and traffic-space attacks. The observation that the inverse mapping from feature space to traffic space is neither invertible nor differentiable constitutes a useful insight with potential implications beyond network security — for instance, in audio processing, NLP with discrete tokens, or medical applications involving tabular data.

The critique of the applicability of L<sub>p</sub>-norms to heterogeneous tabular data is well argued. The authors rightly note that conventional metrics fail to capture inter-feature dependencies and do not correspond to the notion of "imperceptibility" in the context of network traffic.

### Significance

The work lies at the intersection of constrained optimization theory and applied information security, which aligns well with the "Trusted AI" track of the MathAI conference.

The questions raised regarding the convergence of penalty-based methods under a large number of constraints are of fundamental importance for the development of formal robustness guarantees. The empirical observation that 360 constraints in the C-PGD experiment prevented the generation of any valid adversarial examples due to conflicting gradients is particularly telling.

---

## Weaknesses and Limitations

### Survey Methodology

The paper is a survey, and within the conventions of this genre it is competently executed. Nevertheless, the work would benefit from a more transparent methodology for source selection: it would be helpful to specify the inclusion and exclusion criteria for the reviewed works and to delineate the scope of the literature coverage. This is a standard recommendation for survey publications that enhances their value as a reference resource.

### Mathematical Rigor

Formulation (5) for the deep reinforcement learning-based method raises some concerns. In particular, the expression for R<sub>t</sub> employs argmax within the Q-function, which does not fully conform to the standard formulations of Double DQN or Dueling DQN. Clarifying this point would improve the clarity of the exposition.

Furthermore, the authors mention the NP-completeness of the problem in one of the approaches but do not provide a formal analysis of the computational complexity of the methods under discussion. Even a brief assessment of practical tractability would be a welcome addition.

### Analytical Comparison of Methods

For a survey of this scope, it would be natural to offer a more structured analytical comparison of the reviewed methods — for example, in the form of a summary table indicating the type of constraints handled, scalability, theoretical guarantees, and experimentally demonstrated effectiveness based on data from the primary sources. This would facilitate the reader's selection of an appropriate approach for a given task.

### Connection to Defensive Mechanisms

Given that the work is addressed to the Trusted AI community, a more detailed discussion of how understanding constrained adversarial attacks can inform the development of more robust defense mechanisms would be valuable. In particular, the relationship to certified defense methods merits dedicated consideration.

---

## Questions for the Authors

**Question 1.** Have you considered the applicability of certified defense methods, such as randomized smoothing, to the network traffic domain? What specific challenges arise in this setting?

**Question 2.** Section 5 notes that many IDS rely on classical ML models (e.g., Random Forest) with non-differentiable loss functions. What alternative attack methods, beyond substitute models, do you consider promising for such targets?

**Question 3.** You present a compelling critique of L<sub>p</sub>-metrics but do not propose alternatives. What type of distance metric do you consider appropriate for network traffic — a weighted feature combination, expert-defined distances, or some other approach?

**Question 4.** Given that the complete incorporation of all domain constraints is acknowledged to be infeasible, do you propose any criterion of "sufficiency" for constraint coverage in practical applications?

**Question 5.** How do you relate your work to the field of neuro-symbolic AI? Could the integration of domain constraints be viewed as a form of incorporating symbolic knowledge into the neural optimization process?

---

## Overall Recommendation

**Weak Accept (Accept with Minor Revisions).**

The paper constitutes a solid survey of the constrained adversarial attack problem in the network traffic domain and can serve as a useful entry point for researchers new to this area. The originality rating reflects the inherent characteristics of the survey genre rather than any deficiency in research effort. The positive recommendation is motivated by the relevance of the topic to the MathAI conference, the quality of the systematization, and the practical utility of the identified open problems.

For the final version, the authors are encouraged to:

- clarify mathematical Formulation (5);
- include a discussion of the relationship to certified defense methods;
- describe the methodology used for literature selection;
- provide a summary table offering an analytical comparison of the reviewed methods.

---

### Decision · Program_Chairs · 2026-03-14

**Decision:**

Accept (Oral)

**Comment:**

Dear Author(s),

On behalf of the Program Committee of the International Conference on Mathematics of Artificial Intelligence (MathAI 2026), we are pleased to inform you that your paper has been accepted for an oral presentation at MathAI 2026.

Your paper was evaluated through a rigorous two-stage review process involving both automated screening and expert review by members of the Program Committee. The reviewers recognized the quality and contribution of your work.

Presentation details:

- Format: Oral presentation (15–20 minutes + 5 minutes Q&A)
- Mode: You may present either in person (offline) at the conference venue in Sirius, Russia, or remotely via Zoom. Please indicate your preferred mode when confirming your participation.
- Conference dates: Marh 30 - April 3, 2026
- Website: https://mathai.club

Next steps:

1. Please confirm your participation and presentation mode by replying to this email mathai.club@yandex.ru no later than March 15, 2026 18:00 Moscow time.
2. If you plan to attend in person, the organizing committee will provide accommodation details separately.
3. Please prepare your final camera-ready manuscript according to the formatting guidelines available at https://mathai.club and upload it to OpenReview by March 15, 2026 18:00 Moscow time.

Should you have any questions regarding the program, logistics, or your presentation slot, please do not hesitate to contact us.

We look forward to your contribution to MathAI 2026.

With kind regards,

MathAI 2026 Program Committee
International Conference on Mathematics of Artificial Intelligence
https://mathai.club
OpenReview: https://openreview.net/group?id=mathai.club/MathAI/2026/Conference
Telegram: https://t.me/MathAI_club
Email: mathai.club@yandex.ru